# Role of DSCAM in the Development of Neural Control of Movement and Locomotion

**DOI:** 10.3390/ijms22168511

**Published:** 2021-08-07

**Authors:** Maxime Lemieux, Louise Thiry, Olivier D. Laflamme, Frédéric Bretzner

**Affiliations:** 1Centre de Recherche du Centre Hospitalier Universitaire de Québec, CHUL-Neurosciences P09800, 2705 boul. Laurier, Québec, QC G1V 4G2, Canada; maxime.lemieux.1@ulaval.ca (M.L.); louise.thiry@mail.mcgill.ca (L.T.); olivierdlaflamme@hotmail.com (O.D.L.); 2Department of Psychiatry and Neurosciences, Faculty of Medicine, Université Laval, Québec, QC G1V 4G2, Canada

**Keywords:** DSCAM, mouse genetics, gait, posture, spinal cord, motor cortex, motor control, locomotion, neurophysiology, neuroanatomy

## Abstract

Locomotion results in an alternance of flexor and extensor muscles between left and right limbs generated by motoneurons that are controlled by the spinal interneuronal circuit. This spinal locomotor circuit is modulated by sensory afferents, which relay proprioceptive and cutaneous inputs that inform the spatial position of limbs in space and potential contacts with our environment respectively, but also by supraspinal descending commands of the brain that allow us to navigate in complex environments, avoid obstacles, chase prey, or flee predators. Although signaling pathways are important in the establishment and maintenance of motor circuits, the role of DSCAM, a cell adherence molecule associated with Down syndrome, has only recently been investigated in the context of motor control and locomotion in the rodent. DSCAM is known to be involved in lamination and delamination, synaptic targeting, axonal guidance, dendritic and cell tiling, axonal fasciculation and branching, programmed cell death, and synaptogenesis, all of which can impact the establishment of motor circuits during development, but also their maintenance through adulthood. We discuss herein how DSCAM is important for proper motor coordination, especially for breathing and locomotion.

## 1. Introduction

Recent mouse studies have investigated the role of Down syndrome cell adhesion molecule (DSCAM) in the development and maintenance of motor and locomotor circuits. In this review, we provide an overview of the functional contribution of DSCAM in regard to other signaling pathways and we detail the role of *Dscam* mutation in the development of the spinal locomotor circuit, generating locomotor pattern and rhythm, the role of primary sensory afferents and propriospinal pathways important for gait and posture, and finally the contribution of supraspinal descending inputs of the brainstem and motor cortex important for voluntary locomotor control in the rodent.

## 2. DSCAM as a Signaling Pathway

Whereas there is an abundant literature regarding the contribution of invertebrate DSCAMs to the formation and maintenance of neural circuits [1,2,3,4], less is known about vertebrate DSCAMs. However, there is more and more evidence that vertebrate DSCAMs play an important role in axonal growth, fasciculation and branching, dendritic arborization, mosaic spacing of cells, and synaptogenesis (for reviews, see [5,6,7,8,9,10]). DSCAMs are widely expressed across the central and peripheral nervous system in vertebrates with 2 isoforms: DSCAM and DSCAML1 [11,12], and this review will focus on the most common isoform: DSCAM (Figure 1A).

### 2.1. Self-Avoidance in Cell and Neurite Spacing

Dendritic and axonal arbors of many neuronal types exhibit self-avoidance, in which branches or neurons can repel each other, thus contributing to cell and neurite spacing (Figure 1B). In invertebrates, the extensive splicing of DSCAM gives rise to thousands of isoforms that contribute to normal dendritic self-avoidance and proper dendritic field organization [13,14,15,16]. Although vertebrate DSCAM does not generate such isoform diversity, it contributes to mosaic spacing of neurons and dendritic tree organization [17,18], but its self-avoidance mechanism seems to be indirect. Indeed, other cell adherence molecules, such as protocadherins, undergo splicing and generate proteins with unique extracellular domains that contribute to extensive combinatorial homophilic interactions, contributing in turn to a similar recognition specificity in dendritic self-avoidance and dendritic field organization [19,20,21,22,23,24,25]. Furthermore, using a series of double mutant mice for DSCAM and different members of the cadherin superfamily [26], it has recently been shown that preventing adhesion can rescue neurite fasciculation in *Dscam*^−/−^ neurons, whereas ectopic expression of cadherins in the absence of DSCAM causes neurite fasciculation, thus arguing that DSCAM, by masking the superfamily cadherin, works indirectly as a self-avoidance cue.

**Figure 1 ijms-22-08511-f001:**
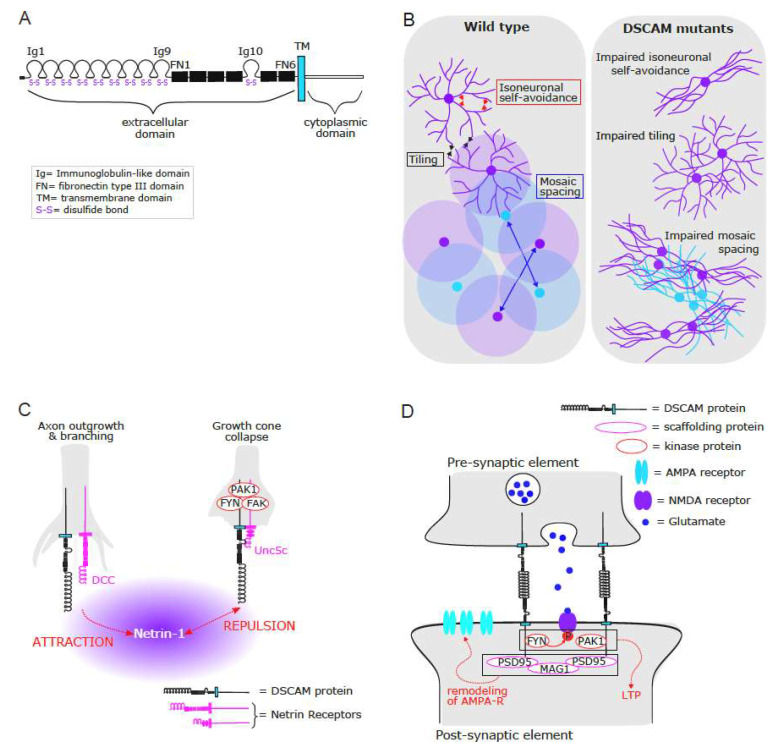
Vertebrate DSCAM protein as an important regulator of cellular and subcellular organization of developing neural circuits. (**A**) Schematic illustrating the structure of vertebrate DSCAM protein. Positioned after a signal peptide (SP), the extracellular portion of the protein is formed by several immunoglobulin-like (Ig) and fibronectin type III (FN) domains. The position of the Ig-10 domain between FN domains 4 and 5 is characteristic of DSCAM proteins. The protein also contains a transmembrane (TM) and a cytoplasmic domain. Each Ig domain contains two cysteines (S) that form an intrachain disulfide bond (Adapted from Montesinos 2014 [7]). (**B**) Schematic illustrating the spacing between two distinct cell types (blue vs. purple) in a wild-type mouse (left panel). DSCAM masks inappropriate adhesion, allowing self-avoidance of neurites from an individual cell (=isoneuronal self-avoidance), leading to neurons occupying spatial domains by arborizing their processes. DSCAM also allows for the complete but nonoverlapping coverage of space by homotypic dendrites (=tiling), thus positioning the soma and establishing a zone within which other neurons of the same type are excluded, spacing their cell bodies (=mosaic spacing). In *Dscam* mutants (right panel), unmasked adhesion drives fasciculation and clustering of a given neuron subtype, resulting in a loss of self-avoidance at the level of both individual cells (isoneuronal self-avoidance) and between cells of a given subtype (tiling and mosaic spacing) (Adapted from Fuerst et al., 2008 [27]). (**C**) Schematic illustrating the role of DSCAM in axon outgrowth and branching (left) or growth cone collapse (right) in response to Netrin-1. DSCAM can cooperate with DCC to induce attraction toward Netrin-1 (left), or with Un5c to induce repulsion in response to Netrin-1 (right). Note that DSCAM and Unc5c physically interact for Netrin binding, contrary to DSCAM and DCC. (**D**) Schematic illustrating the role of DSCAM in glutamate synapse formation, maintenance, and function. Through its interaction with scaffolding proteins such as MAGI and PSD95, DSCAM is involved not only in stabilizing the pre-synaptic element to the post-synaptic, but also in clustering and remodeling of post-synaptic AMPA-like receptors at the post-synaptic membrane during synaptogenesis. Furthermore, the intracellular domain of the DSCAM protein interacts with PAK1 and FYN kinases, which play an important role in long-term potentiation (LTP) and learning.

### 2.2. Developmental Programmed Cell Death Pathway

Cell death markers such as TUNEL and cleaved caspase 3 are decreased in *Dscam* mutant mice and presumably in DSCAM-expressing cells [27,28], contributing to an aberrant cell proliferation and hypertrophy of several regions in the developing brain of *Dscam*^−/−^, *Dscam*^2J^, and *Dscam*^del17^ mutant mice [27,28,29,30]. Moreover, as shown by the increased number of cells in the inner retina of *Dscam* mutant mice, but more modest changes in other layers, this proliferation seems to be cell-specific [17,27,31], thus explaining discrepancies between and within several brain regions upon *Dscam* mutation. 

Furthermore, the number of cells is increased in the *Dscam* and *Bax*^−/−^ mutant retina in absence of the pro-apoptotic *Bax* gene in comparison to *Dscam* mutant or *Bax*^−/−^ mutant mice [32]. Conversely, overexpression of DSCAM decreases the number of cells in the retina, which is rescued in *Bax*^−/−^ and DSCAM-overexpressing mice [32], thus suggesting that DSCAM can promote cell death by acting through Bax and other cell death pathways. However, despite a comparable or lower density in *Bax*^−/−^ mice, cell clustering and spacing are more severely impaired in *Dscam* mutant than in *Bax*^−/−^ mutant retinas, thus arguing that DSCAM, by inhibiting cell death, contributes to some extent but not solely to cell spacing and clustering during development, and presumably through adulthood.

### 2.3. Laminar, Cellular, and Dendritic Organization

Given its role in cell adhesion and repulsion, DSCAM has been proposed to play an important role in the organization of the developing brain [28,32,33,34,35,36,37,38]. *Dscam*^del17^ mutant mice exhibit a prominent and aberrant clumping of neurons in the developing midbrain [28] and retina [17,27,31], thus supporting the hypothesis that DSCAM controls cell spacing. *Dscam* knockdown studies also impair the radial migration of neurons to upper layers by trapping them in the deep layers and intermediate zone of the developing postnatal cortex [38]. Similarly, knockdown studies also prevent the detachment of nascent cells from ventricles in the developing embryonic midbrain [28]. Molecularly, the cytoplasmic domain of DSCAM appears to interact with RapGEF2 or through a protein complex including RapGEF2, MAGI1, and β-catenin, that suppresses the spontaneous activity of Rap1 and attenuates intercellular adhesions with cadherin molecules, thus supporting the hypothesis that DSCAM plays a role in developing laminar and cellular organization. 

As shown in the developing retina [17,27], DSCAM also governs neurite arborization and dendritic self-avoidance at the subcellular level. In the developing cortex, *Dscam*^del17^ mutant mice exhibit transient impairments in the branching of layer V pyramidal neuron dendrites, with an increase in small and immature dendritic spines at the expense of large and stable spines; nevertheless, these changes eventually return to normal through adulthood [34]. As observed in *Xenopus* midbrain neurons [39], DSCAM knockdown also increases the complexity of proximal dendritic branching of in vitro mouse cortical neurons [38], whereas DSCAM overexpression inhibits dendritic branching of in vitro mouse hippocampal neurons [40], thus suggesting that DSCAM overexpression could contribute to the reduced dendritic arborization of cortical and hippocampal neurons of Down syndrome patients [41,42,43]. Taken together, these studies identify DSCAM as an important regulator of the laminar, cellular, and subcellular organization of developing neural circuits. 

### 2.4. Axonal Guidance, Growth, Fasciculation, and Branching

In vitro studies have shown that DSCAM is involved in the regulation of actin cytoskeleton dynamics through its interaction with its ligand Netrin-1 and other Netrin-1 receptors: Unc5c and DCC (Figure 1C). Indeed, whereas DSCAM interacts with Unc5c (uncoordinated 5) to mediate growth cone collapse and repulsion through the assembly of an intracellular signaling complex involving FYN, FAK, and PAK1 [44], DSCAM also collaborates with DCC (deleted in colorectal cancer) to mediate Netrin-1-induced axon outgrowth and attraction [45,46]. Furthermore, DSCAM and DCC colocalize partially with βIII-tubulin on axon branches, Netrin-1 increases this colocalization, and knocking down DSCAM, DCC, or both blocks Netrin-1-induced axonal branching of primary neurons [47], thus suggesting that DSCAM might collaborate with DCC to regulate microtubule dynamics.

Knockdown studies have shown that DSCAM contributes to the axonal growth of dorsal root ganglion cells, cerebellar granular cells, and retinal cells, but also spinal and cortical commissural neurons in developing *Xenopus*, chicks, and mice [38,39,44,45,46]. However, spinal commissural axons appear to be normal in *Dscam*^del17^ mutant mouse embryos [48], as well as in *Dscam* knockdown chick embryos [49], and axons of retinal ganglion cells also properly exit the eye and project normally in the optical tract of *Dscam*^del17^ and *Dscam*^2J^ mutant mice [50], thus suggesting a normal axonal guidance upon *Dscam* mutation. 

Nevertheless, the organization of the axonal growth cone of retinal ganglion cells, and their optic chiasm and tract, are impaired in *Dscam*^del17^ and *Dscam*^2J^ mutant mice [50]. Interestingly, knockdown studies of genetically identified spinal interneurons also show axonal fasciculation defects in chick embryos [49]. Conversely, mutant mice overexpressing DSCAM show an aberrant axonal growth of retinal ganglion cells with a wider ipsilateral optic tract [50], thus supporting a role of DSCAM in axonal growth and fasciculation. 

If cortical neurons project normally through the commissure upon *Dscam* knockdown, the density of axonal commissural branches is decreased in the cortical grey matter of the contralateral cortex [38]. Similarly, anterograde tracing studies have also shown that corticospinal tract axons of the motor cortex of *Dscam*^2J^ mutant mice decussate normally at the level of the medulla and project normally in the contralateral spinal white matter [51], but their axonal terminals exhibit an aberrant branching within the dorsal spinal grey matter. Taken together, these findings suggest that DSCAM contributes to axonal growth, fasciculation, and branching, but not guidance.

### 2.5. Establishment and Maintenance of Synaptic Functions

Given its role in axonal outgrowth [44,45,46] and dendrite morphogenesis [34,38,40], it is not surprising that DSCAM contributes to synaptogenesis and synaptic integration (Figure 1D). DSCAM promotes the targeting of retinal ganglion cell dendrites and bipolar cell axons to the same layer in the retina of chick embryos [52], thus contributing to lamina-specific synaptic circuits. Using *Aplysia* neuronal cultures of sensory neurons and motoneurons [53], it has been shown that pre- and post-synaptic DSCAM are necessary for the normal clustering and remodeling of post-synaptic AMPA-like but not NMDA-like receptors during synaptogenesis, synaptic transmission, and plasticity. Interestingly, NMDA induces local translation of DSCAM in the dendrites of mouse hippocampal neurons [40], as well as in the axonal growth cone [54]. Moreover, the C-termini of DSCAM can interact with scaffolding proteins such as MAGI and PSD95 [55], which could contribute to the remodeling of AMPA-like receptors during synaptogenesis, whereas the intracellular domain of the DSCAM protein interacts with PAK1 and FYN kinases, which play an important role in long-term potentiation and learning [56,57]. Furthermore, the intracellular domain of DSCAM can also interact with IPO5 (a nuclear import protein of the importin β family) that promotes its nuclear translocation. Once in the nucleus, the intracellular domain of DSCAM can regulate the transcription of genes involved in synapse formation and other neuronal processes [58]. Such molecular mechanisms at the membrane and nuclear levels might therefore contribute to the development, maintenance, and plasticity of synapses.

The *Dscam*^2J^ mutant spinal cord exhibits a decrease in the density of glutamatergic pre-synaptic boutons of peripheral sensory afferents onto motoneurons, and a reduced monosynaptic sensorimotor reflex during development and through adulthood [30]. Furthermore, *Dscam*^2J^ mutation also alters the intracortical circuitry of the motor cortex by decreasing the density of intracortical and thalamocortical inputs onto cortical and corticospinal neurons, and by impairing intracortical processing in response to low- and high-frequency stimulation [51], thus preventing short-term plasticity and synaptic integration at the cortical level.

Interestingly, mouse models of Down syndrome that express a third copy of the *Dscam* gene have been instrumental in revealing a wide range of dysfunctions in both glutamatergic and GABAergic synapses in several brain regions, and especially in the hippocampus [59,60,61,62,63,64,65,66,67,68]. Similarly, DSCAM overexpression in *Drosophila* induces synaptic targeting errors [69] and alters glutamatergic synaptic transmission [70], thus arguing that DSCAM is important in synaptic formation, integration, and transmission.

## 3. Posture and Gait in *Dscam* Mutant Mice

### 3.1. Posture

Like *Dscam*^del17^ mutant mice [71], *Dscam*^2J^ mutant mice lacking DSCAM show a hunched posture with limb hyperextension at rest and at slow walking speed (Figure 2A), recapitulating the scaredy cat posture [72], but this hyperextension decreases at high walking speed. Similar to *Dscam*^2J^ and *Dscam*^del17^ mutant mice [71,72], *Dscam* knockdown studies of zebrafish embryos also show a severe phenotype, with a moderate to severe rostro-caudal axis shortening with crooked tail in some embryos [73]. Although no association has been reported in a Chinese Han population [74], genome-wide association studies have previously shown that *CNTNAP2* and *DSCAM* genes are associated with adolescents with idiopathic scoliosis susceptibility in a white population [75], thus suggesting that the hunched posture could be associated with *Dscam* mutation. Although these last studies argue for a genetic origin of the hunched posture upon *Dscam* mutation, the increased motor tone observed in *Dscam*^2J^ mutant mice could be sufficient to promote the hunched posture of *Dscam*^2J^ mice during development.

### 3.2. Intralimb and Interlimb Coordination, and Gait

Gait analysis can be useful in allowing us to investigate the development and establishment of the neural circuit underlying motor control and locomotion. Locomotion is organized in two phases: a swing phase reflecting the recruitment of flexor muscles during which the animal moves its limb forward, and a stance phase supported by the recruitment of extensor muscles during which the animal supports and transfers its body weight forward on its limbs (Figure 2B). Kinematics and electromyographic recordings are currently used to investigate locomotor control: the footfall pattern can be analyzed to monitor interlimb coordination, whereas the angular excursion of limb joints can be analyzed to study intralimb coordination during the swing and stance phase of locomotion. To complement kinematic analyses, electromyographic recordings can detail the spatial and temporal recruitment of muscle activities at a higher resolution than kinematics [76]. Combining data on interlimb coordination and the duty cycle of the stance phase of locomotion, it is then possible to characterize and identify walking vs. running gaits, as well as transition gaits [77].

*Dscam*^2J^ mutant mice exhibit a longer swing phase with hindlimb hyperflexion at the expense of a shorter stance phase during treadmill locomotion at all walking speeds [72]. The increased motor activity in flexor muscles and the overlap in the offset of flexor activity and the onset of extensor activity can contribute to an exaggerated forward placement and a reduced backward placement of the paw, delaying the initiation of the swing phase as well as the transition from the swing to the stance phase.

Although *Dscam*^2J^ mutant mice exhibit hindlimb hyperextension at slow walking speed while the motor drive in the extensor muscle is near normal, they switch to a more normal posture at high walking speed with an increased activity in both flexor and extensor muscles. 

Regarding interlimb coordination and gait (Figure 2C,D), wild-type littermates use typical walking gaits, such as the out-of-phase walk at slow walking speed, which can be regarded as an exploratory gait [77], lateral walk at intermediate walking speeds, and trot at high walking speed, which is the most efficient gait in mice and quadrupeds [77,78,79]. In contrast, *Dscam*^2J^ mutant mice exhibit a very different spectrum of gaits with a higher prevalence of out-of-phase walk at slow and intermediate walking speed, lateral walk instead of trot at high walking speed, and a few episodes of pace at all walking speeds, which is more common in long-legged animals such as horses [80].

Although NCAM mutation has been reported to impair the peripheral nervous system [81,82,83,84], the neuromuscular junction and the contractile properties of muscles and muscle spindles are normal and do not show any signs of motor spasticity or rigidity in *Dscam* mutant mice upon passive limb movements [72], thus supporting the hypothesis that the functional and neurophysiological phenotypes of *Dscam*^2J^ mutants reflect central rather than peripheral neurological changes.

## 4. Spinal Locomotor Circuit

Spinal cord preparations harvested from neonatal rodents have been widely used to investigate the spinal locomotor circuit in the absence of sensory feedback from peripheral and descending inputs from the brain [85,86,87] (Figure 3D). After isolation, the spinal cord is placed in a recording chamber superfused with oxygenated artificial cerebrospinal fluid. Lumbar ventral roots can be attached to suction electrodes to monitor flexor- and extensor-related motoneuronal activities from lumbar L2 and L5 ventral roots, respectively. Bath application of a cocktail mimicking the descending control of the brain or electrical stimulation of dorsal roots can activate the spinal locomotor circuit and generate a locomotor-like activity, with alternation between left and right activities and between flexor- and extensor-related motoneuronal activities [88,89,90,91], which is also called fictive locomotion in the absence of real movements. 

Using these isolated spinal cord preparations [30], it has been shown that *Dscam*^2J^ mutant spinal cords exhibit changes in locomotor pattern and rhythm, with an increased variability in their step-cycle duration (i.e., locomotor frequency) and in the duration of their flexor- and extensor-related activities. Furthermore, whereas the flexor–extensor alternation is normal, *Dscam*^2J^ mutant spinal cord preparations show a decrease in their left–right alternation, with episodes of synchronization, which persists though adulthood [72,92] (Figure 3E).

### 4.1. Genetically Identified Spinal Interneuronal Populations

Based on their transcription factor expression during development, four subclasses of spinal interneurons have been identified in the ventral spinal cord: V0, V1, V2, and V3, which originate from the p0, p1, p2, and p3 progenitor domains, respectively [93,94,95,96] (Figure 3A). Similarly, six subclasses of spinal interneurons also originate from the dorsal progenitors that give rise to dI1, dI2, dI3, dI4, dI5, and dI6 in the dorsal spinal cord [96]. Some of these interneurons are glutamatergic ipsilateral interneurons, such as V2a, V2d, and Hb9, whereas others are GABAergic/glycinergic ipsilateral interneurons, such as V1 and V2b. Others are glutamatergic commissural interneurons (i.e., they project contralaterally on the other side of the spinal cord), such as V3 and V0v, whereas others still are GABAergic/glycinergic contralateral interneurons, such as V0d and dI6 (Figure 3B). Mouse genetic studies have enabled the identification and functional evaluation of six major spinal interneuronal populations (dI6, V0, V1, V2, V3, and Hb9) and several subclasses pertaining to locomotion, that will be presented in the next sections [97,98,99].

### 4.2. Spinal Excitatory Interneurons Important to Generating Locomotor Rhythm

The variability in locomotor rhythm reported in the *Dscam*^2J^ mutant spinal locomotor circuit could reflect changes in network connectivity [30], but also in the intrinsic properties of rhythmogenic interneurons [100]. Among candidates are Hb9 interneurons, excitatory glutamatergic interneurons located next to the central canal of the spinal cord that project ipsilaterally. They can generate an intrinsic rhythm, recapitulating locomotor-like activity in isolated spinal cord preparations, even in the absence of network activity [101,102]. A higher variability in locomotor rhythm has also been reported upon optogenetic or pharmacological manipulation or genetic ablation of other glutamatergic interneuronal subpopulations, including Shox2/Chx10 expressing V2d interneurons [103], Chx10 expressing V2a interneurons [104,105], and glutamatergic commissural interneurons Sim1 + expressing V3 interneurons [106]. Whether intrinsic and extrinsic properties of these excitatory glutamatergic interneuronal subpopulations are impaired in the *Dscam* mutant spinal locomotor circuit remains to be investigated.

### 4.3. Spinal Commissural Interneurons Are Important in Left–Right Coordination

*Dscam*^2J^ mutant spinal cords exhibit an abnormal increase in the number of commissural interneurons [92] (Figure 3F,G). Several signaling pathways contribute to normal development of the spinal circuit, and especially left–right coordination. For example, mutant spinal cords lacking EphA4 or ephrinB3, or their downstream effectors, α2-chimaerin or Nck, exhibit an aberrant increase in the number of commissural interneurons contributing to a left–right synchronization of their locomotor activities [107,108,109,110]. The EphA4-ephrinB3 pathway is thought to prevent ipsilateral interneurons from crossing the midline during normal development by exhibiting a chemo-repulsive signal [111]. Although there is no evidence that DSCAM interacts with the EphA4-ephrinB3 pathway, *Dscam*^2J^ mutant spinal cords exhibit an abnormal increase in the number of commissural interneurons, which could result from an aberrant axonal guidance signal. DSCAM is highly expressed in neurons and axons during embryonic and postnatal development [4,28]. Although knockdown studies using small-interference RNA or blocking their signaling have shown that DSCAM interacts with Netrin-1 in axonal guidance [4,32,45], such a mechanism is likely transient according to *Dscam*^del17^ mutant mouse studies [48]. Alternatively, the aberrant commissural circuit of *Dscam*^2J^ mutant spinal cords could also result from a dysfunctional cell death pathway. Indeed, if DSCAM overexpression increases cell death in the developing retina [32], *Dscam* mutation reduces it in the developing midbrain and retina [28,32], thus arguing that mutation or loss of DSCAM may impair normal apoptosis (i.e., the programmed cell death) during development and contribute indirectly to an excessive survival of spinal commissural interneurons.

### 4.4. Reciprocal Inhibition in Flexor–Extensor Alternation

Spinal cords isolated from neonatal *Dscam*^2J^ mutant mice exhibit an overall normal locomotor pattern and coupling between flexor- and extensor-related motoneuronal activities during neonatal fictive locomotion [30]. However, adult *Dscam*^2J^ mutant mice exhibit hindlimb hyperextension during the stance phase and hindlimb hyperflexion during the swing phase, with a longer stride length and higher stride height [72]. These kinematic changes are associated with an increase in the amplitude and duration of flexor and extensor muscle activities, with episodes of coactivation between flexor and extensor muscles contributing to delay initiation and termination of the swing phase during locomotion in contrast to their wild-type littermates, thus suggesting an improper reciprocal inhibition and recurrent inhibition.

Reciprocal inhibition contributes to a proper timing of activities between the onset and termination of flexor and extensor burst activities through inhibitory Ia interneurons [112,113] (Figure 3C). Recently, mouse studies have identified Engrailed-1 expressing V1 and GATA2/3-expressing V2b interneurons as two inhibitory GABAergic/glycinergic interneurons contributing to reciprocal inhibition and recurrent inhibition [114]. Synaptic silencing of both V1 and V2b interneurons leads to a flexor–extensor synchrony during neonatal fictive locomotion [115], arguing that both V1 and V2b interneurons contribute to flexor–extensor reciprocal inhibition. Furthermore, genetic ablation of V1 interneurons increases the duration of flexor-related activity during neonatal fictive locomotion and induces hyperflexion in the adult mouse [116], thus suggesting that V1 interneurons contribute to proper extension. In contrast, genetic ablation of V2b interneurons increases the duration of extensor-related activity during neonatal fictive locomotion and induces hindlimb hyperextension and delay in the transition from stance to the swing phase through adulthood [116], therefore arguing that V2b interneurons contribute to initiation of the swing phase. Although the activity of V1 and V2b interneuronal circuits is likely normal according to neonatal fictive locomotion studies [30], the coactivation in flexor–extensor activities and the delay in initiation and termination of the swing phase during adult treadmill locomotion [72] suggest that their recruitment might be impaired upon DSCAM mutation.

### 4.5. Motoneuronal Output: Recurrent Inhibition or Excitatory Drive of the Spinal Locomotor Circuit

Adult *Dscam*^2J^ mutant mice also exhibit an increase in the burst amplitude of flexor and extensor muscles during treadmill locomotion [72], and this increased locomotor output could reflect a dysfunction in recurrent inhibition. Indeed, the firing pattern of motoneurons is regulated by recurrent inhibition through Renshaw cells [117,118]. However, there is no change in the density of inhibitory GABAergic boutons onto motoneurons of neonatal and adult spinal *Dscam*^2J^ mice [30], which excludes this interpretation. Nevertheless, there is an increase in the density of excitatory glutamatergic VGluT2 pre-synaptic boutons onto spinal motoneurons of adult *Dscam*^2J^ mice, which suggests that the increased locomotor output results from an increased central excitatory drive of the spinal locomotor circuit.

## 5. Sensory Afferents Modulate Locomotor Pattern

*Dscam*^2J^ spinal motoneurons show a decrease in the density of excitatory glutamatergic VGluT1 pre-synaptic boutons labeling proprioceptive afferents. Peripheral tibial nerve stimulation fails to evoke a consistent and robust H-reflex (i.e., electrical equivalent of the stretch reflex) in adult *Dscam*^2J^ mutant mice. Electrical stimulation of dorsal roots evokes motor responses of smaller amplitude and longer latency, and long trains of electrical stimulation fail to evoke episodes of locomotor-like activity in isolated neonatal *Dscam*^2J^ mutant spinal cord preparations [30]. It remains uncertain if cutaneous, proprioceptive, or both afferents are affected in DSCAM.

### 5.1. Cutaneous Afferents

Cutaneous afferents innervate the skin of the body and relay information in response to skin deformation during locomotion or following contact with an object [119,120]. As shown in the cat [121,122], mechanical stimulation of the dorsum of the mouse hind-paw during the swing phase evokes a stumbling corrective response, recruiting distal flexor muscles to withdraw the paw and step over a virtual obstacle [123]. However, as previously shown by cutaneous denervation in the cat [124,125], genetic silencing of glutamatergic dI3 spinal interneurons that relay cutaneous inputs does not impair overall locomotion in the mouse [126]. Although *Dscam*^2J^ mutant mice exhibit difficulty walking on the rungs of a horizontal ladder or stepping over an obstacle attached to a treadmill belt at slow speed [51], their forelimb steps properly over small and large obstacles at intermediate treadmill speed, thus suggesting that their cutaneous afferents and their integration in the spinal locomotor circuit are normal.

### 5.2. Proprioceptive Afferents

As suggested by their posture at rest and during slow walking gaits [72], proprioceptive afferents of adult *Dscam*^2J^ mutant mice are likely impaired. Proprioceptive afferents relay information about the stretching and loading of muscles during movement and locomotion [127,128]. As previously shown in the cat, stretching the hip flexor muscle in recruiting muscle spindles initiates a swing phase [129,130], whereas loading the extensor muscle in recruiting the Golgi tendon organs delays the initiation of the swing phase [131], thus contributing to the transition from the stance to the swing phase. Furthermore, the position of the hip joint and the hip flexor activity also appear to play an important role in the transition from the swing to the stance phase [132,133].

Removal of proprioceptive feedbacks has shown that they are important in locomotor pattern [134,135]. Adult *Egr3* mutant mice lacking muscle spindles but not Golgi tendon organs can walk properly on a treadmill [134,135], but the precise timing of the offset of ankle flexor muscle activity is perturbed during the swing phase [135] and the speed-dependent amplitude modulation of the ankle extensor is also altered [136]. Interestingly, unloading the sensory feedback from the Golgi tendon organs by challenging *Egr3* mutant mice in a swimming pool induces a synchronization of hip, knee, and ankle flexor activities [135], thus arguing for the importance of both proprioceptive afferents from muscle spindles and Golgi tendon organs to locomotor coordination.

Given that reciprocal and recurrent inhibitions are recruited by primary sensory afferents [118] and that glutamatergic synaptic transmission is impaired upon *Dscam* mutation [53], it is tempting to speculate that a decreased excitatory drive of primary sensory afferents could impair the normal recruitment of V1 and V2b interneurons in *Dscam*^2J^ mutant mice, thus leading to hindlimb hyperflexion, with a higher forward limb placement and trajectory during the swing phase and hindlimb hyperextension during the stance phase of locomotion.

## 6. Forelimb and Hindlimb Locomotor Coordination

Gait analysis can be useful for investigating the development and establishment of motor circuits underlying intralimb and interlimb coordination, as well as forelimb–hindlimb coordination [76,77,87]. It is possible to get insights about the spinal cervical and lumbar locomotor circuits, their reciprocal interaction through propriospinal interneurons, and their modulation by peripheral inputs of primary sensory afferents and by supraspinal descending inputs of the brain upon genetic mutation.

Adult *Dscam*^2J^ mutant mice exhibit a reduced repertoire of gaits with a reduced maximal locomotor speed [92]. In addition to impairing the ability of mice to maintain high treadmill speed with running gaits, the *Dscam* mutation also induces the dominance of lateral walk over trot and the emergence of aberrant gaits, such as diagonal walk and pace, rarely reported in the mouse (Figure 2D). Graph analysis has also shown that gaits of *Dscam*^2J^ mutant mice are less predictable and less stable than normal. This instability in locomotor gait and transition of gaits suggests a reorganization within and between cervical and lumbar locomotor circuits.

### 6.1. Propriospinal Interneuronal Pathways

As previously shown in the cat [137,138,139,140], propriospinal interneurons are important in synchronizing cervical and lumbar motor circuits in various types of locomotion, including stepping, trotting, and swimming. Recently, genetic tracing studies in the mouse have shown that descending glutamatergic propriospinal interneurons of the cervical spinal circuit project primarily contralaterally in the lumbar spinal circuit that controls the hindlimb, whereas descending GABAergic propriospinal interneurons of the cervical spinal circuit project mainly ipsilaterally in the lumbar circuit [141]. Conversely, ascending glutamatergic propriospinal interneurons project contralaterally in the cervical circuit, whereas ascending GABAergic propriospinal interneurons project ipsilaterally in the cervical circuit, thus genetically identifying reciprocal propriospinal interneuronal pathways connecting cervical and lumbar spinal locomotor circuits important to interlimb coordination. Among several classes of genetically identified interneurons, intersectional genetics has revealed that V0 commissural interneurons and Shox2-expressing V2 ipsilateral interneurons are identified as propriospinal interneurons [141].

### 6.2. Functional Role of Propriospinal Pathways in Forelimb–Hindlimb Coordination

Interestingly, mutant mice lacking excitatory glutamatergic V0v interneurons can walk and gallop, though they cannot trot [79,142]. Genetic ablation of descending lumbar-projecting cervical propriospinal interneurons leads to a synchronization of left–right hindlimbs at intermediate treadmill speed [141], thus recapitulating the phenotype of V0v mutant mice. The predominance of lateral walk over trot in *Dscam*^2J^ mutant mice suggests therefore that V0 propriospinal interneurons could be impaired and prevent a normal synchronization of the diagonal coupling between the opposite forelimb and hindlimb that secures the trot.

Furthermore, given the role of excitatory glutamatergic Shox2-expressing V2 interneurons in rhythm generation [103], V2 propriospinal interneurons could sustain rhythm generation from the brainstem to the cervical and lumbar spinal circuits during locomotion. In support of that hypothesis, descending cervical propriospinal interneurons relay inputs of sensory afferents from the forelimb and neck, but more importantly from the head as well [141,143,144]. Interestingly, genetic ablation of descending propriospinal interneurons induces a decrease in travelled distance and maximal locomotor speed in the mouse [141]. Given the inability of *Dscam*^2J^ mutant mice to maintain locomotion and generate running gaits at high treadmill speed, *Dscam* mutation could impair the normal propagation of the locomotor rhythm from the brainstem to the spinal cord through a descending chain of V2 propriospinal interneurons. 

## 7. The Brainstem

Given its contribution to basic motor controls such as walking and breathing, the brainstem is crucial in the normal development of motor control. However, little is known about the role of DSCAM, as is the case for other signaling pathways in the functional organization of brainstem motor circuits.

### 7.1. Locomotor Brainstem Circuits

As discussed in the previous section, DSCAM exhibits locomotor deficits that are central in origin and might include a reorganization of brainstem networks. As recently described [76,145,146,147], the medullary reticular formation is organized into discrete nuclei, with the gigantocellular reticular nucleus located along the rostrocaudal axis of the medulla and the lateral paragigantocellular nucleus located ventrally and laterally to the gigantocellular reticular nucleus in the caudal medulla (Figure 4A). Both nuclei appear to integrate and relay the command of supraspinal locomotor centers, such as the Mesencephalic Locomotor Region to the spinal locomotor circuit, through their reticulospinal pathways.

#### 7.1.1. The Lateral Paragigantocellular Nucleus (LPGi)

Using a photo-activator, it has been shown that long trains of photo-stimulation delivered above glutamatergic neurons of the lateral paragigantocellular nucleus can initiate locomotion and set locomotor speed in response to an increase in laser intensity [148]. In contrast, genetic ablation of this neuronal population abolishes locomotion evoked by photo-stimulation of glutamatergic neurons of the Mesencephalic Locomotor Region, thus arguing that glutamatergic neurons of the lateral paragigantocellular nucleus relay glutamatergic supraspinal locomotor inputs and likely glutamatergic inputs of the cuneiform nucleus, which is known to initiate and accelerate locomotion [149,150,151]. Given the difficulty of *Dscam*^2J^ mutant mice to initiate locomotion and use standard walking gaits, such as lateral walk and trot at slow and intermediate treadmill speeds, but also to use running gaits such as gallop and bound at high treadmill speed [72,92], the connectivity between supraspinal locomotor centers and lateral paragigantocellular pathways is likely impaired upon *Dscam* mutation.

#### 7.1.2. The Gigantocellular Reticular Nucleus (Gi)

Using transcription factors that regulate gene expression during development, it has been shown that Lhx3 and Chx10 expressing brainstem neurons, also called V2a neurons, exhibit a tonic firing pattern and are activated after an episode of locomotion or upon electrical stimulation of supraspinal locomotor centers [152]. Using a photo-activator, bilateral photo-stimulation of V2a neurons of the gigantocellular reticular nucleus stops locomotion, whereas their pharmacoinhibition increases locomotor activity [153], thus suggesting that these V2a gigantocellular reticular neurons are stop cells. Although most V2a gigantocellular reticular neurons are glutamatergic [152,153], not all glutamatergic gigantocellular reticular neurons are necessarily V2a. Nevertheless, long pulses of photo-stimulation delivered above glutamatergic gigantocellular reticular neurons increase step-cycle duration by inducing a coactivation in flexor–extensor hindlimb muscles ipsilateral to the stimulation site during locomotion [154]. This increase of the stance duration and the delay in the onset of the next swing phase suggest that both glutamatergic and V2a gigantocellular reticular neurons can reset and stop locomotor rhythm. 

In line with that suggestion, the coactivation in flexor and extensor hindlimb muscles of *Dscam*^2J^ mutant mice and their difficulty in generating a normal spectrum of gaits could reflect an increased drive from the glutamatergic gigantocellular reticular nucleus. However, electrical stimulation of the *Dscam*^2J^ mutant gigantocellular reticular nucleus evokes short-latency motor responses of normal amplitude, duration, and latency [51], thus arguing that reticulospinal efficacy is normal upon *Dscam*^2J^ mutation and instead favors neural changes at the spinal level. 

### 7.2. Respiratory Brainstem Circuits

The brainstem is also the locus of neural networks controlling breathing. Interestingly, *Dscam* mutant mice exhibit irregular respiration and lower ventilatory response to hypercapnia [29,155]. Three functional oscillatory networks have been identified to date as key components from the caudal to the rostral medulla: the pre-Bötzinger complex, the post-inspiratory complex, and the retrotrapezoid nucleus/parafacial respiratory group (Figure 4A,D). These networks are rhythmogenic (i.e., they can generate their own rhythm independently), they depend on synaptic glutamatergic transmission, and they are functionally coupled (e.g., they are reciprocally connected).

The pre-Bötzinger complex is a functional region located in the caudal medulla that contains glutamatergic and GABAergic neurons linked to inspiration and pre-inspiration [156,157]. Mouse genetics studies have shown that Dbx1-expressing neurons within the pre-Bötzinger complex are rhythmically active [158,159]. Their photoactivation resets the breathing rhythm generated by the pre-Bötzinger complex [160,161], while their ablation stops or disrupts the rhythm [162]. The post-inspiratory complex is another group located rostrally to the nucleus ambiguus, which contains glutamatergic and cholinergic neurons that contribute to inspiratory/expiratory transition [163]. Finally, the retrotrapezoid nucleus/parafacial respiratory group is located ventrolaterally to the facial nucleus and contains glutamatergic neurons and contributes to active expiration and central chemoreception of CO_2_ and pH, important during aerobic exercise [164,165,166,167]. Some glutamatergic neurons within this third group express the transcription factor Phox2b during development, but their rhythmogenic or chemoreceptive functions are still unknown [168,169,170]. As recently reviewed [171,172,173], rhythmogenic activity in each complex results from three features: a recurrent synaptic excitation that contributes to synchronizing respiratory neurons, an intrinsic bursting conductance of pacemaker cells that enhance spiking and amplify synchronization, and a concurrent inhibition that regulates synchronization and increases variability in rhythm. This rhythmic synchronization hence contributes to producing robust and dynamic breathing in regard to aerobic demand and context.

Using *Dscam* mutant mice, plethysmographic recordings have shown that mutant mice exhibit irregular respiration and lower ventilatory response to hypercapnia [29,155] (Figure 4B,C), thus arguing for functional impairments of the chemosensory and respiratory retrotrapezoid nucleus/parafacial respiratory group. Furthermore, using brainstem–spinal cord preparations isolated from neonatal mice, electroneurographic recordings have also shown that the activity of motoneurons controlling the diaphragm shows an irregular rhythm with frequent apneas upon *Dscam* mutation, and voltage-sensitive dye imaging has shown that pre-inspiratory neurons lose their normal synchronization in *Dscam* mutant brainstem preparations, thus suggesting functional impairments of the intrinsic and/or synaptic excitation of pre-inspiratory and inspiratory pre-Bötzinger neurons. In summary, these data suggest that both the inspiratory pre-Bötzinger complex and the expiratory and chemoreceptive retrotrapezoid nucleus/parafacial respiratory group are impaired upon *Dscam* mutation.

### 7.3. Coupling between Locomotion and Respiration

Respiration and locomotion can be loosely coupled during slow walking gaits, while they tend to be synchronized with a 1:1 coupling ratio during fast trot and running gaits at high speeds [174,175]. Running gaits, such as gallops and especially bounds in quadrupeds, contribute to a forward movement of the viscera that increases thoracic volume and facilitates inspiration during the aerial phase of the animal, while a backward movement of the viscera in compressing the thoracic cage promotes expiration during the landing phase [174]. Furthermore, passive limb movements can increase ventilation rhythm [176], and electrical stimulation of sensory afferents or locomotor activity of the lumbar spinal circuit in the absence of real movements can also increase respiratory rhythm using brainstem–spinal cord preparations isolated from neonatal rodents [177,178,179]. Finally, supraspinal locomotor centers can also increase locomotor and respiratory rhythm in decerebrate animals, but also in decerebrate and paralyzed animals in the absence of sensory feedback [180]. In summary, sensory afferents from moving limbs, mechanoreceptors of the lungs, locomotor activity, as well as supraspinal locomotor centers can all modulate the locomotor-respiratory coupling to adjust aerobic demand to the context, such as exploration for food foraging, chasing prey, or fleeing a predator.

As mentioned in a previous section, glutamatergic V2a neurons of the gigantocellular reticular nucleus fire tonically, are important to locomotion, and project in the spinal cord [152,153,181,182], but they also project in the inspiratory pre-Bötzinger complex [183]. Interestingly, in addition to impairing locomotor rhythm, genetic ablation of V2a neurons also decreases breathing rhythm, which can be normalized by increasing neuronal excitability via pharmacological drugs. Nevertheless, decreased activity of the pre-Bötzinger complex eventually leads to postnatal death, such as in *Dscam*^−/−^-null mutant mice [29,155], thus raising questions about a concomitant or sequential functional impairment of respiratory and locomotor networks upon DSCAM mutation during development.

## 8. The Motor Cortex and Its Corticospinal Pathway

In addition to exhibiting a reduced repertoire of locomotor gaits over a wide range of treadmill speeds [72,92], adult *Dscam*^2J^ mutant mice show voluntary motor control impairments while walking on the rungs of a horizontal ladder or while stepping over an obstacle during treadmill locomotion [51], thus arguing for neurological changes within the motor cortex and its corticospinal tract. The motor cortex exhibits cortical representations of the whole body, with territories specifically dedicated to the control of the arm or the leg, for example [184,185,186,187,188,189,190]. Within a cortical representation, the motor cortex is organized in 6 layers, with upper layers containing cortical neurons integrating and processing thalamic and cortical inputs from other brain regions and deeper layers in which cortical neurons project to other regions, such as corticospinal neurons projecting in the spinal cord [191,192,193,194,195].

### 8.1. Development of the Corticospinal Tract

Axonal guidance signals are important for axonal growth and lateralization of corticospinal tract axons in the contralateral spinal cord. Several signaling molecules, such as Semaphorin-6A and its Plexin-A4 receptor [196], as well as cell adhesion molecules L1 [197], NCAM [198], and Netrin-1 and its DCC and Unc5h3 receptors [199,200], contribute to the normal decussation of the corticospinal tract at the junction of the medulla and the spinal cord (i.e., pyramidal decussation). For example, *Dcc*^kanga^ mutant mice that survive through adulthood exhibit an aberrant pyramidal decussation of their corticospinal tract that mis-projects ipsilaterally in the spinal cord, in addition to exhibiting functional impairments while walking on a horizontal ladder [199,200], thus arguing for the contribution of DCC in normal pyramidal decussation of corticospinal tract axons. Despite in vitro evidence of transient interactions of DSCAM with Netrin-1 or its receptors [44,45,46,47,48,49,201], corticospinal tract axons of *Dscam*^2J^ mutant mice do not exhibit any obvious defects in their pyramidal decussation [51] (Figure 5A). Furthermore, corticospinal tract axons of *Dscam*^2J^ mutant mice also exhibit normal lateralization of their axons within the spinal grey matter, and stimulation of the motor cortex evokes normal motor responses in the contralateral side of their body [51].

Although pyramidal decussation and lateralization of corticospinal axons in the spinal cord are normal upon *Dscam*^2J^ mutation, these axons present an aberrant wider dorsoventral projection with a more dorsal component within the spinal grey matter [51] (Figure 5B). As recently reported in the retinogeniculate pathways [50], this wider dorsoventral projection points toward potential deficits in axonal fasciculation and growth of corticospinal tract axons. Although cortical forelimb and hindlimb representations are normal in *Dscam*^2J^ mutant mice, the motor threshold for evoking movements is four-fold higher than in their wild-type littermates [51]. This reduced corticospinal efficacy argues for a decreased excitability of cortical and/or spinal motor networks.

Interestingly, unilateral inactivation of the motor cortex by intracortical muscimol infusion permanently impairs development of the normal dorsoventral distribution of corticospinal terminals during the postnatal period in wild-type mice [202], thus recapitulating the pattern of *Dscam*^2J^ corticospinal terminals. As previously shown in the developing retina [32], *Dscam* mutation could preclude the normal refinement/pruning of corticospinal axonal terminals by preventing programmed cell death and promoting aberrant synaptic contacts within the dorsal spinal cord grey matter during development.

### 8.2. Functional Organization within the Motor Cortex

If intracortical micro-stimulation reveals a reduced corticospinal efficacy [51], electrical stimulation of the pyramidal tract shows a normal efficacy and excitability of corticospinal tract axons in *Dscam*^2J^ mutant mice, thus arguing for decreased cortical and intracortical excitability within the *Dscam*^2J^ mutant motor cortex (Figure 5C,D). Although *Dscam*^del17^ mutation only transiently impairs dendritic arborization and spine formation of the developing motor cortex [34], *Dscam*^2J^ mutation permanently impairs the density of synaptic inputs through adulthood [51]. Indeed, *Dscam*^2J^ mutation decreases the density of VGluT2-expressing thalamocortical inputs relaying sensory feedback and increases the density of VGluT1-expressing cortical inputs on cortical interneurons and corticospinal tract neurons [51] (Figure 5F), whereas it spares the density of inhibitory inputs. Physiologically, despite a normal spontaneous rhythmic cortical activity, *Dscam*^2J^ mutation impairs intracortical connectivity between cortical sites in response to single pulses of intracortical micro-stimulation and decreases synaptic integration in response to single pulses and trains of intracortical micro-stimulation (Figure 5E), thus arguing for dysfunctions in temporal facilitation and short-term plasticity. Given its role in synaptic targeting, stabilization of dendritic spines, and the functional organization of the pre- and post-synaptic compartments of excitatory glutamatergic transmission in various neural circuits and animal models [30,34,39,52,53,69], DSCAM likely contributes to normal functional organization of the motor cortex and its corticospinal drive, important for voluntary motor control.

## 9. Conclusions

Through its multiple roles in axonal guidance, branching, and fasciculation, as well as in dendritic spine formation and stabilization, DSCAM contributes to the formation of the spinal locomotor circuit, its peripheral sensory inputs, and its propriospinal pathways, which impact spinal locomotor functions during development and the spectrum of gait and posture through adulthood. Moreover, DSCAM appears to contribute to normal development of brainstem respiratory networks, which could condition the functional organization of motor and locomotor circuits during postnatal development given the coupling between respiratory and locomotor networks. Furthermore, DSCAM is also important in voluntary locomotor control by promoting synaptic integration and short-term plasticity within the motor cortex. With the emergence of new tools in mouse genetics, it will be important to further investigate the conditional knockdown or overexpression of DSCAM in genetically identified neuronal subpopulations of the spinal cord, peripheral afferents, brainstem, or motor cortex, to gain a better understanding of the role of DSCAM in the formation and maintenance of motor circuits important to stereotypic and voluntary motor control and locomotion.

## Figures and Tables

**Figure 2 ijms-22-08511-f002:**
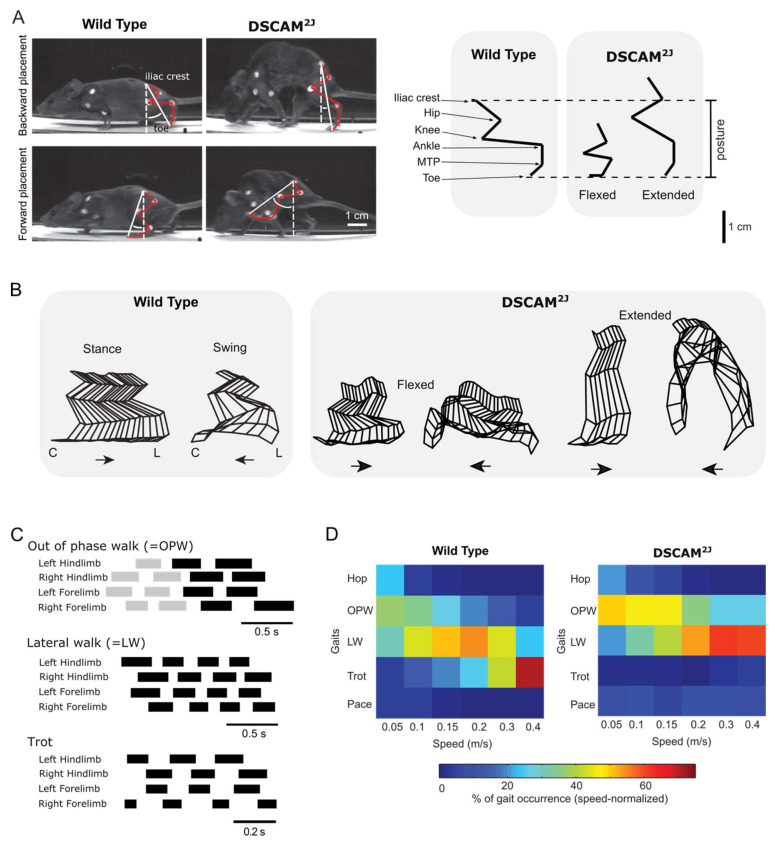
Posture and gait in *Dscam*^2J^ mutant mice. (**A**) Left panel: photographs (left panel) of wild-type and *Dscam*^2J^ mutant mice at their maximal backward (top) and forward limb placement (bottom). Note the aberrant forward limb placement of *Dscam*^2J^ mutant mice. Right panel: stick diagrams illustrating the extended posture of *Dscam*^2J^ mice observed at low walking treadmill speed (0.1 m/s). Posture was measured as the vertical distance between the 4th digit and the iliac crest, before onset of the swing phase. (**B**) Stick diagrams of the hindlimb during the stance and swing phases at low walking treadmill speed. Note the occurrence of 2 postures for *Dscam*^2J^ mutant mice (flexed vs. extended). (**C**) Gait diagrams illustrating typical locomotor gaits identified at walking speeds: out-of-phase walk (OPW), lateral walk (LW), and trot. Black bars represent the stance phase of the gait, and gaps represent the swing phase. Grey bars in OPW are another type of gait. (**D**) Color-coded matrix of gait occurrence (in %) in wild-type (left) and *Dscam*^2J^ mutant (right) mice for each walking speed. Note in wild-type (left) a change from out-of-phase walk to lateral walk, and then from lateral walk to trot. Changes are delayed in *Dscam*^2J^ (right) and trot is almost absent (Adapted from Lemieux et al., 2016 [72]).

**Figure 3 ijms-22-08511-f003:**
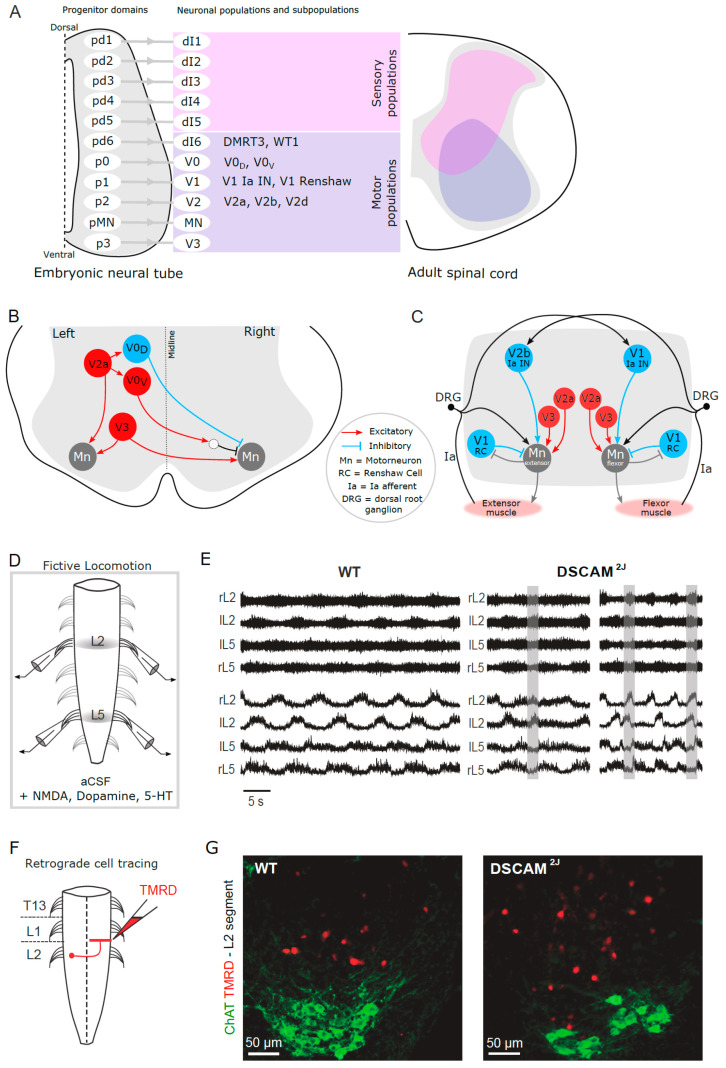
The spinal locomotor circuit in *Dscam*^2J^ mutant mice. (**A**) Schematic illustrating the progenitor domains that give rise to genetically distinct spinal interneuronal populations along the dorsoventral axis of the neural tube. Sensory interneurons, shown in pink, are located in the dorsal part of the adult spinal cord, while motor interneurons, shown in purple, are located in the ventral spinal cord. (**B**,**C**) Schematics summarizing the genetically identified interneuronal populations contributing to left–right (**B**) or flexor–extensor (**C**) coordination, with their neurotransmitter phenotype and their network connectivity. (**D**) Schematic illustrating the isolated spinal cord preparation in a recording chamber superfused with oxygenated artificial cerebrospinal fluid (aCSF). Rostral and caudal ventral roots can be attached to suction electrodes to monitor flexor- (L2) and extensor (L5)-related electroneurographic (ENG) locomotor-like activity upon bath application of a pharmacological cocktail mimicking the descending brain command. (**E**) Representative examples of ENG activities recorded from L2 and L5 lumbar ventral roots of one wild-type (left) and two *Dscam*^2J^ mutant (right) mouse spinal cords. Top traces correspond to the raw ENGs, and bottom traces to the integrated ENGs. Note the synchronous ENG activity of the right and left L2 ventral roots in the *Dscam*^2J^ spinal cords (indicated by shaded areas). rL2 = right L2 ventral root; lL2 = left L2 ventral root; lL5 = left L5 ventral root; rL5 = right L5 ventral root. (**F**) Ventral view of the spinal cord illustrating the protocol used to label commissural interneurons (CINs) in the L2 segment. (**G**) Representative examples of tetramethylrhodamine (TMRD+)-labeled CINs (red) and choline acetyltransferase (ChAT+) motoneurons (MNs; green) in the ventral horn of the L2 segment from wild-type and *Dscam*^2J^ neonatal spinal cords. Note the aberrant number of CINs identified in the *Dscam*^2J^ spinal cord (Adapted from Thiry et al., 2016 [30]).

**Figure 4 ijms-22-08511-f004:**
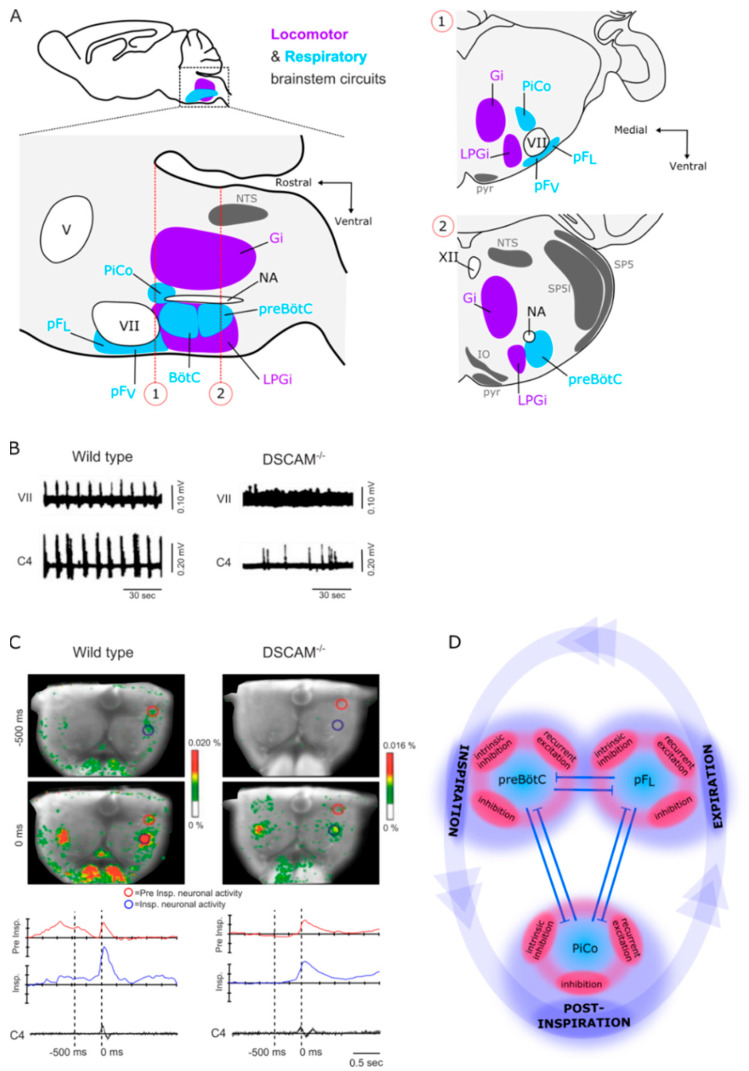
Locomotor and respiratory brainstem networks likely impaired in *Dscam* mutant mice. (**A**) Parasagittal view of the brainstem containing the locomotor (purple) and respiratory (blue) brainstem networks. Brainstem nuclei associated with integration and relay of the command of the supraspinal locomotor centers are shown in purple: the gigantocellular reticular nucleus (Gi) and the lateral paragigantocellular nucleus (LPGi). Brainstem sites associated with breathing motor pattern or sensorimotor integration are shown in blue: the pre-Bötzinger complex (preBötC; inspiratory), the pFL (expiratory), and the more medial chemo-sensitive ventral parafacial respiratory group (pFV; rhythmogenic in the perinatal period only), as well as the “post-inspiratory complex” (PiCo; hypothesized to underlie post-inspiration) and the expiratory Bötzinger complex (BötC). Insets 1 and 2 show transverse sections at the level of the pF and PiCo (dotted line 1) and preBötC (dotted line 2). Cranial motor nuclei controlling airway resistance muscles, the hypoglossal motor nucleus (XII) and the nucleus ambiguus (NA), as well as facial muscles, the facial motor nucleus (VII), and the trigeminal motor nucleus (V), are shown in white. The nucleus of the solitary tract (NTS), the spinal trigeminal tract (SP5), the spinal trigeminal sensory nucleus interpolaris (SP5I), the inferior olive (IO), and the pyramidal tract (pyr) are shown in grey. (**B**) Representative example of the output of the VII nerve and the C4 ventral root activities in medulla–spinal cord preparations of newborn wild-type and *Dscam*^−/−^ mice. Note how the VII nerve activities for *Dscam*^−/−^ are tonic (Adapted from Amano et al., 2009 [29], Copyright 2009 Society for Neuroscience). (**C**) Optical images showing the average of 50 respiratory cycles triggered by C4 inspiratory activity in wild-type (left) and *Dscam*^−/−^ mutant (right) mice. Images were recorded 500 ms (top pictures) or 0 ms (bottom pictures) before C4 inspiratory burst and are superimposed on the ventral surface of the medulla. The red circles indicate the most prominent area of Pre-I neuronal activity, and the blue circles indicate that of Insp neuronal activity. Representative examples of optical responses within red and blue circles (78 pixels) are quantified and shown as red and blue traces respectively, below the optical images (bottom panel). Note that the Pre-I neuron activity in *Dscam*^−/−^ mice disappeared (Adapted from Amano et al., 2009 [29], Copyright 2009 Society for Neuroscience). (**D**) Inspiration, post-inspiration, and expiration constitute a continuous unitary breathing cycle relying on the rhythmic synchronization of the three respiratory rhythmogenic structures illustrated in (**A**).

**Figure 5 ijms-22-08511-f005:**
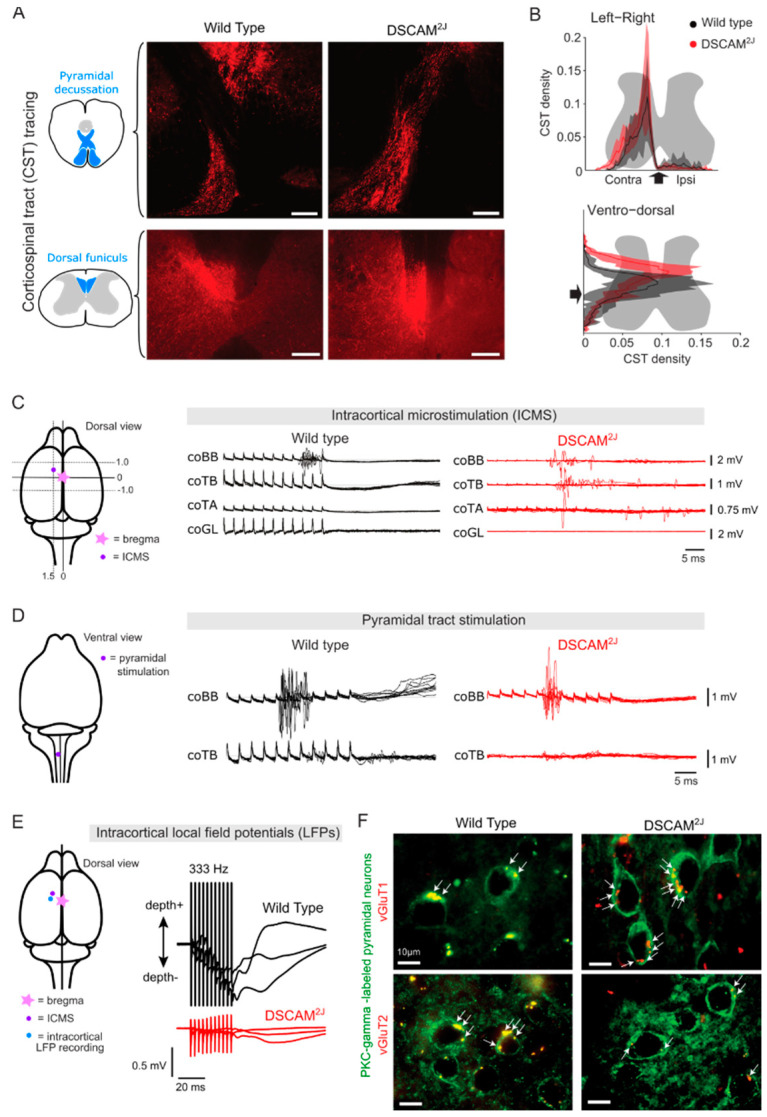
Motor cortex and corticospinal tract in *Dscam*^2J^ mutant mice. (**A**) Representative examples of BDA-labeled corticospinal tract axons at the level of the pyramidal decussation (top panel) or through the dorsal funiculus (bottom panel) in wild-type (left panel pictures) and *Dscam*^2J^ (right panel pictures) spinal cords. (**B**) Axonal density of BDA-labeled corticospinal tract axons as a function of the medio-lateral axis (top) and dorso-ventral axis (bottom) of the spinal cord (Mean (thick line) ± SD (area); *n* = 6 WT and 5 *Dscam*^2J^ mice). The projection of corticospinal terminals in *Dscam*^2J^ (red) is more dorsal compared to the normal mediolateral projection in wild-type (black). (**C**) Schematic illustrating the site of intracortical micro-stimulation (ICMS; purple dot) according to the Bregma (star along the midline), and representative examples of electromyographic (EMG) responses evoked by the ICMS in a wild-type (black) and a *Dscam*^2J^ mutant mouse (red). (**D**) Schematic illustrating the site of pyramidal tract stimulation (purple dot), and representative examples of EMG responses evoked by stimulation of the pyramidal tract in a wild-type (black) and a *Dscam*^2J^ mutant mouse (red). (**E**) Schematic illustrating the site of ICMS (purple dot) as well as a site of local field potential (LFP) recordings (blue dot) in the motor cortex, and examples of LFP recordings during trains of pulses at 333 Hz in a wild-type (black) and a *Dscam*^2J^ mutant mouse (red). (**F**) PKC-gamma-labeled pyramidal neurons (green) and immunochemistry of vGluT1 (top panel, red) or vGuT2 (bottom panel, red) in the motor cortex of wild-type and *Dscam*^2J^ mutant mice. vGluT1+ or vGluT2+ boutons are indicated by arrows (Adapted from Laflamme et al., 2019 [51]).

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
