# Peer review of "Role of DSCAM in the Development of Neural Control of Movement and Locomotion"

_ijms, 2021, doi:10.3390/ijms22168511_

Round 1

Reviewer 1 Report

In this review, the authors outlined some key information about the cell adhesion molecules associated with Dawn Syndrome (DSCAM) as a signalling molecule influencing the structural neuronal organization and signalling pathways, including cell death signalling. A more detailed focus has been on the DSCAM role in locomotion across different regions of the nervous system. This area of research is interesting and important as DSCAM appears to exert multifunctional roles, having a range of various impacts on multiple pathways. I support the publication of this manuscript pending some minor improvements.

  • There is a lack of basic information about the DSCAM molecular biology and mechanism(s) of actions. It would be interesting and useful for a wide audience to understand how a general expression profile for DSCAM looks like in central neurons? Does it act intracellularly or by binding to surface receptors after being released through vesicle mechanisms in order to regulate neuronal guidance? Same for structural development.
  • Some signalling mechanism(s) suggested would provide better clarity for the DSCAM involvement into a tiny regulation of structural plasticity - any involvement of binding protein microfilaments established vs trophic effects?
  • Some typos are present throughout the text.

Author Response

We thank the reviewer for his/her feedback on our review.

  • There is a lack of basic information about the DSCAM molecular biology and mechanism(s) of actions. It would be interesting and useful for a wide audience to understand how a general expression profile for DSCAM looks like in central neurons? Does it act intracellularly or by binding to surface receptors after being released through vesicle mechanisms in order to regulate neuronal guidance? Same for structural development.
  • Some signalling mechanism(s) suggested would provide better clarity for the DSCAM involvement into a tiny regulation of structural plasticity - any involvement of binding protein microfilaments established vs trophic effects?

As suggested by the reviewer, we have included additional references about DSCAM molecular biology and mechanisms of action in Section 2, which include the binding of DSCAM to ligands and other receptors and the intracellular mechanisms regulating neuronal growth. We have also included more molecular information regarding the interaction of DSCAM with protein microfilaments.

  • Some typos are present throughout the text.

The article has been proofread. 

Reviewer 2 Report

The manuscript: "Role of DSCAM in the development of the neural control of movement and locomotion “by Maxime Lemieux and colleagues review the available literatures investigating the role of DSCAM in the development and maintenance of motor and locomotor circuits. The authors have abundantly discussed the clinical, pathological and molecular aspects of DSCAM with apt references. After thoroughly going through the manuscript, I have a couple of minor comments:

  1. The abstract is abruptly ended. I would suggest the authors to include a sentence or two that would reflect the significance of this manuscript. What would this manuscript cater to the readers?
  2. Please mention the elongated form of DSCAM : Down syndrome cell adhesion molecule, when this term is mentioned for the first time in the manuscript.
  3. The review is based on different studies performed in rodents. Hence this information should be clearly reflected in the abstract and Introduction sections.

Author Response

We thank the reviewer for his/her feedback on our review.

  1. The abstract is abruptly ended. I would suggest the authors to include a sentence or two that would reflect the significance of this manuscript. What would this manuscript cater to the readers?

We added a sentence to highlight the significance of this manuscript.

2. Please mention the elongated form of DSCAM : Down syndrome cell adhesion molecule, when this term is mentioned for the first time in the manuscript.

We made changes in the abstract and in the first sentence of the introduction.

3. The review is based on different studies performed in rodents. Hence this information should be clearly reflected in the abstract and Introduction sections.

We added this information in the abstract and the introduction.